# Dynamic Epigenetic Changes during a Relapse and Recovery Cycle in Myalgic Encephalomyelitis/Chronic Fatigue Syndrome

**DOI:** 10.3390/ijms231911852

**Published:** 2022-10-06

**Authors:** Amber M. Helliwell, Peter A. Stockwell, Christina D. Edgar, Aniruddha Chatterjee, Warren P. Tate

**Affiliations:** 1Department of Biochemistry, School of Biomedical Sciences, University of Otago, Dunedin 9016, New Zealand; 2Department of Pathology, Dunedin School of Medicine, University of Otago, Dunedin 9016, New Zealand

**Keywords:** ME/CFS, DNA methylation, RRBS, DMAP, Epigenetics

## Abstract

Myalgic Encephalomyelitis/Chronic Fatigue Syndrome (ME/CFS) is a complex disease with variable severity. Patients experience frequent relapses where symptoms increase in severity, leaving them with a marked reduction in quality of life. Previous work has investigated molecular differences between ME/CFS patients and healthy controls, but not the dynamic changes specific to each individual patient. We applied precision medicine here to map genomic changes in two selected ME/CFS patients through a period that contained a relapse recovery cycle. DNA was isolated from two patients and a healthy age/gender matched control at regular intervals and captured the patient relapse in each case. Reduced representation DNA methylation sequencing profiles were obtained spanning the relapse recovery cycle. Both patients showed a significantly larger methylome variability (10–20-fold) through the period of sampling compared with the control. During the relapse, changes in the methylome profiles of the two patients were detected in regulatory-active regions of the genome that were associated, respectively, with 157 and 127 downstream genes, indicating disturbed metabolic, immune and inflammatory functions. Severe health relapses in the ME/CFS patients resulted in functionally important changes in their DNA methylomes that, while differing between the two patients, led to very similar compromised physiology. DNA methylation as a signature of disease variability in ongoing ME/CFS may have practical applications for strategies to decrease relapse frequency.

## 1. Introduction

Myalgic Encephalomyelitis/Chronic Fatigue Syndrome (ME/CFS) is a lifelong severely debilitating disease from which only a small proportion of individuals eventually fully recover (<5%) [1]. While currently not well understood, it is estimated to have a global prevalence of ~1% [2] and to have a higher overall disease burden than conditions like multiple sclerosis, autism or HIV/AIDS [3]. Patients experience a wide variety of debilitating symptoms including severe fatigue, post exertional malaise, and cognitive, sleep and orthostatic dysfunctions [4]. These symptoms vary in severity such that ~25% of patients are house or bedbound throughout the illness. The remaining 75% of those affected transition to a life-long chronic phase where they may be able to participate in work and hobbies, albeit with a reduced capacity. However, they are vulnerable to frequent debilitating “relapses”, particularly after even minor stress. 

The disease presentation and key research to date indicate that there is a complex pathophysiology affecting ME/CFS patients, with biological functions reduced in a number of systems including immune/inflammatory, and neurological as well as in metabolism. For example, an analysis found 80% of 612 metabolites analysed in plasma of ME/CFS patients were significantly decreased, indicating that there was an overall reduction in metabolic activity in patients. This has been compared to the ‘dauer effect’, a shut down like hibernation in animals [5]. Additionally, patient mitochondria have a dysfunction in the mitochondrial complex V where the energy molecule ATP is synthesised, resulting in proteins in the upstream mitochondrial complexes, particularly complex 1, being up regulated as an apparent attempt to compensate [6], as well the metabolic pathways and mechanisms to control regulation of reactive oxygen species [7]. This could explain one component of why ME/CFS patients are unable to respond biologically to day-to-day stresses, let alone high-level stress events. We have proposed a theory based on fluctuating neuroinflammation to explain the sustained chronic state of the illness and ‘relapse recovery’ cycles [8]. It hypothesises that neuroinflammation of the hypothalamus’ stress centre within the paraventricular nucleus could be responsible for the unexplained prolonged and fluctuating symptom presentations [8]. More recently we have developed a model to explain the molecular mechanisms of neuroinflammation that sustain disease and promote relapses in ME/CFS and Long COVID [9].

The absence of a specific molecular diagnostic test, and also the fluctuating variations across patients in disease presentation and symptom severity, has made understanding ME/CFS and conclusions from molecular data difficult. When studies focus on large cohorts of patients, they often include very varied presentations of ME/CFS, and different studies have often used different clinical case definitions for diagnosis. A recent study of the involvement of cytokines addressed the diversity within the patient cohorts on a molecular scale, and found that a large number of pro-inflammatory cytokines were found to be linearly associated with ME/CFS severity [10]. This investigation highlighted the dilemma of many investigations targeting ME/CFS, since the cytokines associated with severity were not sufficient to distinguish the patients from the controls because of the abundance of mild ME/CFS cases in the study group. Other studies have approached this issue by classifying ME/CFS patients into different subgroups. A study in 2008 examining expression levels of transcripts classified 7 subtypes, through mean relative transcript quantities with 88 transcripts that corresponded with clinical severity [11]. In recent years, DNA methylation has been applied to investigate the disease status of ME/CFS patients, and a number of studies have found important differences separating the patients from controls [12,13,14,15,16,17]. A recent publication with this technology has identified four subtypes utilising DNA methylation and symptom severity [18], with key differentially methylated genes between subtypes having primarily immune and metabolic functions. This study indicated that molecular analyses could differentiate patients with the molecular changes reflecting physiology relevant to the observed symptoms.

The gradual trend to a more personalised approach taken by these investigations is an important step towards understanding the intricacies of ME/CFS. Many patients, once they have entered the chronic state of their disease following an initial acute period often lasting several years, experience frequent extreme symptom fluctuations characteristic of a relapse event. No published molecular studies have yet followed patients longitudinally through a ‘relapse recovery’ cycle. In order to understand in depth a disease as complex as ME/CFS this more personalized approach with individual patients is both informative and appropriate, not only for researchers in their studies but also for patients in the management of their disease. Indeed, precision medicine is becoming more readily accessible not only as a research tool to understand the impact of disease on an individual, but also how they will respond to a specific medical intervention [19,20]. This seems especially relevant for the study of ME/CFS where patients have a wide-ranging level of functionality, for example in their ability to exercise, their cognitive deficits, and often different comorbidities.

How can precision medicine be applied to ME/CFS? DNA methylation is an important epigenetic modification that affects the expression of genes without altering the genomic code itself. This specific analysis is with increasing precision helping researchers to bridge the gap between understanding genetic risk and assessing environmental contributions to disease. Changes are captured that are not permanently reflected in the genome but occur in individuals as a result of the disease. An excellent example of this is an in-depth investigation that followed 87 individuals that had transitioned from a pre-diabetic state to a diabetic state. It revealed methylation differences that occurred before the switch to the disease state [21]. Since DNA methylation can capture transcriptional changes that reflect physiological variations, it is an ideal tool in ME/CFS to determine temporal changes in genomic regions that reflect the symptom fluctuations. These are not so easily detectable or may not yet be present in single time point proteome or transcriptome analyses. Understanding variation in an individual’s dynamic epigenetic code with sampling over a precise time period time can provide an insight into the molecular activity and course of their disease.

## 2. Results

### 2.1. Study Design and Participants

Blood was taken from two ME/CFS patients and a healthy age matched control at 5 spaced time points spanning an eleven-month period that captured a health relapse in the ME/CFS patients from their typical compromised health state (see Figure 1A). The participants gave a subjective numerical assessment on a scale of 1–10 as an indicator of their relative health [22]. As is shown in Figure 1A patient 1 (hereafter referred to as P1) showed a drop from a relatively good health state, self-ranked as ‘7’; ‘well’—at the first sample time **a** to an off scale ‘−3’ and ‘−2’, ‘fragile’ indicating a severe relapse condition at time points **b** and **c**. She then showed relative recovery to a health status ‘7’ again at sample times **d** and **e**. Patient 2 (hereafter referred to as P2) in contrast had a more fragile steady state ‘4–6’, mainly ‘fragile’—across the five separate sample collection points with a drop into a relapse ‘2’ at the time of sample **c**. The control remained in excellent health ‘10’ throughout the timeline. The term ‘recovery’ is used in this investigation to define the time points following the relapse where the individual returns to the state of health they experienced prior to the relapse. It is also used as the term, for convenience, to define the time point(s) prior to the relapse event.

The data from Reduced Representation Bisulphite Sequencing (RRBS) of each of the 5 samples from the three subjects were analysed using the Differential Methylation Analysis Package (DMAP) platform with intra-individual variably methylated fragments (iVMFs) identified for each patient and the control (Figure 1B) -named ME-iVMFs. Fragments suitable for analysis had 10 or more reads of at least 2 CpG sites. A total of 13,954, 53,442 and 38,135 met these criteria for P1, P2 and the C respectively. The statistical parameters applied to the analysis are described in methods. For the control individual, there were a total of 6324 fragments that met the significance threshold FDR < 0.05. The qualifying fragments had a median size of 80 bp and an average size of ~84 bp. These fragments contained 52,791 CpG sites, with an average of 8.4 CpGs in each fragment. For P1, there were a total of 2788 statistically significant variably methylated fragments with a total of 22,550 CpG sites and an average of 8.1 per fragment (FDR corrected *p* < 0.05). These fragments themselves had a mean length of ~78 bp with a median length of ~75 bp. For P2, 11,577 fragments with a total of 87,734 CpGs and an average of 7.6 CpGs per fragment met the same significance threshold, and had a median fragment size of 75 bp, and an average fragment size ~80 bp. 

### 2.2. Dynamic Analysis of DNA Methylation Variation

Initially the variation in the methylomes across the time points within each individual was investigated by performing a comparison of the variability at each CpG site. This was calculated from the number of statistically significantly differently methylated CpG sites (*p* < 0.05, methylation difference > 15% compared with the other time points) that were unique to each time point. The percentage of unique DNA methylation variations within each sample was derived from this number compared with the total number of CpG sites analysed for that individual. For example, in the healthy control at time point A 1276 statistically significantly differentially methylated CpGs were identified from a total of 119,931 CpGs compared with the other four time points giving a unique variation of 0.12% (see Figure 2A). This was a consistent pattern, with each time point of the control having a similarly low level of unique differential methylation at CpG sites (ranging from 0.12–0.15%). By contrast the patients showed about a 20-fold greater variation -P1, at time point **a** compared to their other four time points had 2.06% of sites differentially methylated uniquely, and P2 at time point **a** had a similar level of variation of 2.67%). The unique differential methylation at the five time points ranged from 2.06–3.78% in P1 and 1.91–2.67% in P2.

Further predictions of variability based on the DMAP-produced fragment methylation data on further analysis showed that the patients were again more variable than the control, though not as distinctly obvious as from the individual CpG site methylation comparisons described above. The number of statistically significant variably fragments for each individual was divided by the total number of fragments assessed for that individual to produce a variability score. For C, P1, and P2, respectively the overall variability scores were 0.17, 0.20, and 0.22. In order to assess the relative variability of the two patients and the control across functionally important regions of the genome their individual data were extracted across regions of interest, such as within gene bodies, Transcriptional Start Sites (TSS) upstream regions of 10,000 bp (see Figure 2B), and relative CpG island regions of <500 bp with more than 55% GC content position) (Figure 2C).

As seen in Figure 2B the healthy control showed a lower level of variability compared to both the patients at almost every gene related site analysed apart from the TSS where all three individuals in the analysis show similar levels of variability. P2 with the more fragile health showed a higher level of variability compared to both the control and P1.

As seen in Figure 2C the variability of the three individuals across the CpG island related features reflected the same pattern as in Figure 2B at the gene related regions. The control showed the lower variability score compared to both patients. As before, P2, who had the more debilitating ongoing ME/CFS, showed consistently higher levels of variability across the features analysed when compared to P1.

### 2.3. Common ME-iVMFs Methylation Patterns in Patients

Continued analysis of ME-iVMFs included only those fragments that were present in all 15 samples of the three individuals, which resulted in a total of 577 common fragments. Figure 3 is a heatmap that shows the methylation variation across these fragments. It identified the hierarchical clustering of the individual samples based on methylation percent values at each segment, with the associated dendrogram in Figure 3A clearly showing the relationships between the 15 individual samples. The variation within each patient and within the control sample (**a** to **e** time point samples) is lower than the variation among the samples (P1, P2, C), since the heatmap and associated dendrogram clearly grouped the 15 samples into three separate groups that relate to each individual (P1, P2, C). However, it also shows visually the variation within each individual with the 5 different time points clearly showing differences in methylation at a number of fragments.

Initial investigations of these 577 common fragments involved comparing the two patients individually with the matched control in a differential methylation analysis. If the mean methylation difference was greater than 15% between the patient and control methylation scores at these fragments, they were then investigated further. Figure 3B shows the differentially methylated fragments across 68 such selected fragments for P1 vs. C and Figure 3C shows the 53 selected for P2 vs. C.

To further analyse the data for the differentially methylated fragments in the patients compared with the control that fell within gene bodies, STRING.org pathway enrichment analysis was performed. Of the 26 genes that contained differentially methylated fragments in P1 compared with C, two pathways were identified; Nicotine addiction and Morphine addiction due to the presence of genes GNAS, CACNA1A and GABRD. The 23 genes that contained differentially methylated fragments in P2 compared with C showed two pathways with the protein domains; Transforming growth factor-beta (TGF-beta) family and Immunoglobulin C-2 Type, that were identified from the genes; IGSF9B, OPCML, GDF7, CERS1 and LINGO3. 

### 2.4. Identifying Methylation Pattern Associated with the Relapse Condition

In order to find fragments with changes relevant to the relapse in the patients, the data from the overall 577 ME-iVMFs were correlated with the patients self-reported health scores. A Pearson’s correlation coefficient was calculated using the association between the methylation percent at each fragment to the individuals self-reported heath score (as seen in Figure 1A). A minimum Pearson’s correlation coefficient of 0.9 was set. In order to further filter the fragments and select those that reflect the greatest changes in methylation between the patients self-assessed ‘relapse’ and better health ‘recovery’ conditions, a methylation difference was calculated based on the average mean methylation percentages of the relapse and recovery samples, for example, for P1 time points “**b**” and “**c**” were ‘relapse’ and “**a**”, “**d**” and “**e**” were classified as ‘recovery’. A minimum differential methylation of +/− 15% was set. This correlation analysis for P1 identified 17 fragments (Table 1). A total of 14 fragments were identified using this method for P2 (Table 2).

P1-**a** to P1-**e** in Table 1 represents percent methylation data from the samples taken from P1 at each of the time points **a** to **e**. As can be seen at time points **b** and **c** during which there was a self-reported severe relapse (Figure 1A) there was a much lower methylation rate (number italicised) than in the samples from the ‘recovery’ times **a**, **d** and **e**.

Whereas the ME-iVMFs that associated with the relapse condition of P1 were all hypomethylated (see Table 1), for P2, while the majority of the fragments 1–11 were also hypomethylated, three (12–14) by contrast were hypermethylated (shown in bold in Table 2) in the relapse condition (P2-c) compared to the recovery conditions. 

The fragments for both patients show a number of interactions with genomic elements including direct overlaps with gene bodies as well as regulatory elements as recorded in GeneHancer, and USCS genome browser recorded clusters of regulatory interactions between regulatory elements and gene bodies. As these fragments show clear changes in the methylation state of the individual across their relapse and recovery states it has important implications on the regulatory behaviour of a number of associated genes. 

### 2.5. Relapse Associated Methylation Signature Exhibits Striking Variation Compared to Control

Figure 4 gives examples of the top 6 fragments across the five time points for both patients that had the greatest level of differential methylation between their relapse and recovery states. Figure 4 clearly shows that there are clear changes in methylation within the two patients at the relapse condition. In Figure 4A the relapse condition is shown for P1 at time points **b** & **c** compared to recovery time points **a**, **d** & **e**, and in Figure 4B for P2 with relapse at time point **c** and **a, b, d** & **e** for recovery. Due to their genomic location, these fragments have important functional implications, for example from P1 shown in Figure 4A is a fragment that is located within the first intron of NLRP7 gene. It also overlaps with an enhancer (GH19J054952) and directly overlaps a region of regulatory interaction for NLPR2 in addition to being located within a region of Dnase hypersensitivity. As previously mentioned all these fragments were hypomethylated in the relapse state of P1, illustrated in the examples shown in Figure 4A, indicating that the corresponding regulatory features likely have a downstream up-regulation on associated genes.

Both hypomethylation and hypermethylation is shown in Figure 4B with the examples from P2. P2 also has a number of fragments of regulatory importance such as a fragment located on chr1:155098923-155098964 that is located within an archived promoter region (GH01J155123) and has 37 target genes. Additionally, this fragment overlaps with a number of clustered interactions between Genehancer regulatory elements and genes for; *DAP3, CLK2, DMP3, GBAP1, THBS3, EFNA1*. Another fragment of interest shown in Figure 4B is located within the 17th (last) exon of SEMA6B, as it encodes a protein that may be involved in both peripheral and central nervous system development. Additionally, this fragment overlaps with a Genehancer archived promoter region (GH19J004539). It overlaps with a DNAse hypersensitivity cluster and four clustered interactions of Genehancer regulatory elements and genes (*YJU2, PLIN5, SEMA6B* and *LRG1*).

In order to investigate the potential impact of the methylation variation on the patient’s molecular activity during relapse recovery cycles a total gene list was built that associated with the regulatory elements found to be overlapping with the fragments described in Table 1 and Table 2. There were a total of 157 genes associated with the 17 ME-iVMFs identified on relapse in P1, and 127 genes associated with the 14 ME-iVMFs identified in P2 (see Appendix A ‘Genes associated with P1′ and ‘Genes associated with P2′).

### 2.6. Simulated Relapses for the Control Subject Identified Fewer Variable Methylated Genes Than the Patients

To determine whether the ME-iVMFs and associated genes were actually due to the ME/CFS relapse and recovery conditions and not simply a result of random chance and random methylation variation, the control sample was also analysed in two separate determinations as though the healthy control also had the relapse health scores of the two patients, respectively, at the appropriate time points. Thus, each patient relapse health scores were assigned ‘artificially’ to the relevant control time points in separate analyses to calculate the correlation and determine differential methylation from the control data between these simulated ‘relapse’ and ‘recovery’ states. From these analyses, 11 fragments with only 39 genes associated with them (see Appendix A ‘Control_Filt Correl with P1′ and, see Appendix A ‘Genes_assoc with C-P1 condition’) met the filtering requirements with P1′s health scores, and 14 fragments with 53 genes with P2′s health scores (see Appendix A ‘Control_Filt Correl with P2′ and, see Appendix A ‘Genes_assoc with C-P2 condition’). This compares with 157 genes in the data comparison and health states comparison for P1 and 127 genes in the analysis for P2. To determine whether the relationship between the number of associated genes identified per fragment was significantly higher in the patients relapse times, the number of genes identified from each fragment from these ‘simulated’ control analyses and the ‘real’ patient analyses were subjected to an unpaired t-test. The number of genes per fragment were significantly higher for the patient group compared to the two control simulations, with a *p*-value = 0.0053 (see Figure 5A). This implies that while a proportion of the methylation changes seen in the patients may be due to random methylation variation, most of the identified changes during relapse in the patients are due to their physiological relapsed state and are associated with important regulatory regions of the genome linked to ME/CFS disease presentation.

The potential functional associations of each gene that were linked to the variably methylated fragments associated with a ‘relapse’ event were determined. The gene functions identified by this analysis associated with patient symptom fluctuations indicate a change primarily in ‘immune response’ for both patients. Additional functions associated with the genes involved metabolism and transcription for both patients. P1 also had a number of genes involved in cell cycle progression while P2 had a larger number of neuronal related genes. Gene annotations for each gene are listed in the Appendix A ‘Genes associated with P1’ and ‘Genes associated with P2’. Of the immune related genes identified, a number were associated with activities implying increases in the inflammatory response in individuals, with specific functions linked to NF-kappa B activity, wound healing, cytokine release, and angiogenesis observed multiple times. This suggests that during a period of relapse the patient’s immune systems are in an enhanced inflammatory state compared to their relative ‘recovery’ periods. 

## 3. Discussion

Previous studies [12,13,14,15,16,17], including our own study that described the first DNA methylome of ME/CFS patients produced by Reduced Representation Bisulphite Sequencing (RRBS) [16], have established ME/CFS patients at a fixed time point in their disease display an altered DNA methylome in comparison to matched controls. This current study is the first of its kind to analyse the DNA methylome of ME/CFS in individual patients across a longitudinal timeline to investigate a change in health status. Utilising the principles of precision medicine it has identified two key features: (i) the number of variably methylated sites and fragments of the genome are much greater in the two ME/CFS patients than in the control at each time point of the longitudinal study and, (ii) the severity of ME/CFS symptoms during a relapse is associated with methylation variation at key genomic features. The variable methylated DNA fragments enabled us to identify statistically important features specifically associated with a significant ‘relapse’ in the health of the two patients, compared with their prior health and their recovery after the relapse. The genomic features implicated regulatory changes affecting primarily immune functions with associated inflammation, but also metabolic, neurological and mitochondrial functions in patients as they experience symptom fluctuations along the course of their disease.

### 3.1. Benefits of DNA Methylation for a Precision Investigation of ME/CFS

The individuals selected to participate in this study were within a similar weight, age range with the same gender, ethnicity and lifestyles. This was done to prevent any potential confounding factors, since DNA methylation is a dynamic epigenetic mark known to vary due to environmental factors [23,24]. These patients had taken part in many of our studies, had been diagnosed by the same expert ME clinician and we had detailed information on comorbidities, medications, and general lifestyle, family status, including their diets, and level of activity. They were carefully selected to be as closely matched as possible and that their diets were similar and healthily balanced with the main nutritional groups. Indeed they had no comorbidities, had not experienced a pregnancy, but both had taken melatonin to help with sleep, and one patient was taking medication for controlling blood pressure. They were both originally paediatric cases having contracted the illness in their teens and had had their illness formally diagnosed for 6 and 10 years, respectively, but it is likely to have started earlier than that. Hence, The specific criteria for patient selection utilised aimed to ensure that the variation in methylation would be primarily due to fluctuations in ME/CFS symptom severities. 

While previous epigenetic studies have utilised primarily array-based methods, studies involving the same method of RRBS and analysis platforms described here have been performed successfully previously [25,26]. Indeed, our recent study [16] with this method with ME/CFS patients gave methylation changes that significantly overlapped with the other similar studies of this disease that used the array technology [12,13,14,15,17]. Utilising RRBS technology a large number of changes were identified that differentiated ME/CFS patients from controls. The use of RRBS here has followed extensive in-house development and experience with the platform used [25,26,27]. The advantage of using RRBS is that it identifies changes not captured by the array-based studies as it is not limited to the set number of sites in the array, allowing wider coverage of the whole genome.

DNA methylation is an excellent method to investigate physiological changes as it is reflective of transcriptional changes linked to the disease state, and so is very appropriate to study the relapse recovery cycle of ME/CFS. DNA methylation is a versatile method to investigate an individual’s physiology [21]. Small observed changes often reflect much larger changes occurring in a subpopulation of cells that are obscured by the broader range and number of cells from which the DNA is taken. Notably, previous research has indicated that even small measured methylation changes can have large impacts on the associated expression levels of a gene [28]. For example, a recent investigation found that even a small change in methylation percent of (1%) was associated with a two-fold change in expression of insulin like growth factor-2 (IGF2) [29]. A key study relevant to our ‘relapse and recovery’ in ME/CFS patients showed DNA methylation to be changed in at risk individuals before their transition to diabetes [21]. This application of personalised medicine allows DNA methylation variation to be utilised not only to distinguish patients from healthy controls, but also to provide a more specific pathophysiological understanding of an individual patient’s disease trajectory.

### 3.2. Inter-Individual Differences Indicate Increased Epigenetic Variation Linked to Disease Severity

As we develop a deeper understanding of the onset of ME/CFS, it is becoming clear that there is an underlying genetic predisposition in combination with an environmental trigger to precipitate an altered homeostatic state or compromised health ‘baseline’ in patients [30]. Once the disease progresses past the initial acute stage ~75% of patients can transition to a chronic state but the partial recovery is interspersed with frequent periods of relapse followed by relative recovery to the initial compromised health state again. This ‘new normal’ chronic state of ME/CFS for patients may leave them more vulnerable to even minor changes in their environment that would not affect a healthy person. For ME/CFS patients in their altered homeostatic state a dramatic change in physiological state can easily be precipitated.

Initial analysis of the DNA methylation of the genome-wide CpGs of the patients and the healthy control in this study supported the idea that ME/CFS patients are more vulnerable to environmental changes. Month to month, the unique variability in methylation in the healthy control, who had stable excellent health throughout the longitudinal study, was low <1 per 500 sites, but both of the patients had a much higher level of unique variability at each sampling time point at between 1 per 20–50 sites. A similar estimate of variability was performed utilising the DMAP fragment methylation each containing multiple sites, and produced similar but less dramatic trends with the patients having 0.20 and 0.22 variability scores compared with 0.17 for the healthy control (based on the number of statistically significant variable fragments divided by the overall number of fragments in each individual). A key determinant however, is not the extent of variation but the variation at regions of functional importance across the genome, such as in proximity to CpG islands (often associated with regulatory regions), and upstream of and within gene bodies. This investigation found that the patients were consistently more variable than the control at all regions investigated (Figure 2) with the exception of the Transcriptional Start Site (TSS) where both patients and control had similar levels of variability. Of importance to note is that, while both patients were much more variable than the control, P2, in a more compromised state of health throughout the longitudinal timeline, was consistently the more variable of the two patients. The results from this study indicate that not only are patients more epigenetically variable than a healthy control, but also illness severity may be positively associated with methylation variation. 

### 3.3. Intra-Individual Variation Identifies Regulatory Regions

An individual is often their own best control for personalised medical applications, especially in studies like this where there is fluctuating health during a longitudinal disease course. Individuals have fluctuating baseline DNA methylation, so important changes occurring within an individual could be obscured when compared to a control [31]. In this analysis however, when the fifteen samples were clustered based on the methylation scores of the 577 significant fragments identified in all samples (Figure 3), while intra-individual sample variation was indeed revealed, the heatmap and associated dendrogram produced by hierarchical clustering showed that the inter-individual variation clearly differentiated the three individuals and was greater than this intra-individual variation.

For this reason, while a healthy control was included in the analysis, the ‘relapse recovery’ study focused primarily on variation within each individual patient and within the control as three separate individuals, utilising each as their own ‘control’ along a longitudinal time scale (for example Figure 4). Variably methylated fragments were identified in both the patients that strongly associated with the individuals self-reported health scores (r > 0.9) with a distinct methylation percentage difference between the ‘relapse’ and ‘recovery’ conditions (+/– 15%). These thresholds enabled us to capture the more relevant changes occurring in the DNA methylation as a result of the relapse condition as discussed above since even small changes in methylation are often are indicative of larger transcriptomic changes. The control was also analysed in the same manner by ‘simulating’ a relapse and analysing time point samples **b** & **c** (as though it were a relapse as experienced by P1), and sample **c** (as in P2). This determined how many variably methylated fragments are likely to associate, by chance alone independent of disease, when the patient health scores are arbitrarily assigned to the control. A number of variably methylated fragments were identified and were further investigated to identify any functional associations. However, it was clear that the downstream gene associations were much lower when compared to the two patients during relapse (as shown in Figure 5). 

From the variably methylated fragments identified in the patients a large number of downstream genes were associated through either direct physical overlap with the variable fragment, association with a promoter or enhancer, or within a region of regulatory interaction as recorded on UCSC genome browser. They were functionally relevant to physiological changes occurring in the patients as they experience fluctuations in health in a ‘relapse’ and ‘recovery’ cycle. The large majority of the intra-individual variable methylated fragments (ME-iVMFs) were hypomethylated in the relapse condition compared to the recovery condition (only three hypermethylated from P2) (Table 1 and Table 2) indicating that there would be a corresponding increase in transcription in the downstream genes associated with the regulatory features. As there are such a large number of genes associated with the ME-iVMFs identified in this investigation it suggests there are consequentially wide-ranging regulatory changes occurring in patients.

### 3.4. Immune and Inflammatory Changes Implicated in Relapse-Recovery Cycle

While there was a broad range of functional roles identified that were performed by the genes associated with the significant ME-iVMFs (see Appendix A ‘Genes associated with P1′ and ‘Genes associated with P2′) the largest category identified encompassed genes involved immune/inflammatory functions, then in metabolic pathways. As these biological systems have been implicated from previous ME/CFS research studies [6,32,33] it was not surprising that such functional categories would be highlighted as ME/CFS patients experienced fluctuations in their health.

The immune functions identified have important functional relevance to the presentation of ME/CFS. In their relapse compared to their recovery states, P2 had with 34 immune related genes affected. Of these genes, *CXCR2* and *CXCR1* indicated the potential activation of the interleukin-8-mediated signalling pathway. *IL8* has already been observed as the gene most differentially expressed between ME/CFS and controls [7]. Other previous studies also have observed a significantly higher level of IL-8 in severely affected ME/CFS patient group compared to both healthy controls and moderately affected ME/CFS patients [33].

P1 also showed a number of affected genes that like *IL8* are known to be associated with inflammatory responses, for example, *NLRP7*, and genes associated with NF-kappa-B function (*COMMD5, LRRC14, TONSL*). P2 additionally also showed a similar relationship with a number of the immune related genes having inflammatory roles including (*TICAM1 and IL17RA*) which are involved in the positive regulation of cytokine production in inflammatory responses. Significantly a number of the immune related genes from P2 are associated specifically with inflammatory disorders including genes involved in the neutrophil degranulation pathway including; *TMBIM1, SLC11A1, MOSPD2, CRCR2, CRCR1* and *LRG1*.

Among the additional genes of interest identified during relapse in the ME/CFS patients were seven mitochondrial genes in P1 that included *ACOT9*, which is a member of the acyl-CoA family involved in the hydrolysis of Coenzyme A. *HADHA, HADHB* are both involved in mitochondrial beta-oxidation of long chain fatty acids into either 3-etoacyl-CoA if NAD is present, or acetyl CoA if both NAD and coenzyme A are present [34]. In ME/CFS patients it has been hypothesised that a number of factors may be interfering with the production of coenzyme A as a result of inflammation, and with reactive oxygen species through the pyruvate dehydrogenase kinase pathway in the mitochondria [35]. As the activity of mitochondrial beta-oxidation is key to cellular energy production, P1 may be showing the effects of reduced mitochondrial function in the relapse condition that reflects the severity of her relapse state.

While the majority of previous work investigating DNA methylation in ME/CFS patients has focused on the differences between the patients and healthy controls, a 2018 study classified patients into 4 subgroups based on DNA methylation patterns associated with symptom severity [18]. DNA methylation from 1939 genomic sites was utilised as a signature to differentiate the four subgroups. Of these, the top differentially methylated sites had associations related to immune signalling. The subtypes of ME/CFS with the more severe symptom presentation in terms of post exertional malaise were the sites with the highest differential methylation indicating changes in metabolic and immune responses. When considered together with the outcomes of this current study, where the relapse events also highlighted regions potentially affecting the function of immune, inflammatory and metabolic activity, it reinforces the importance of fully understanding the dysfunction of these pathways, not only in patients compared to healthy controls but in individual patients along their disease course.

This investigation has shown regulatory disruptions occurring in the patients associated with their self-reported relapse events. It is worth noting that, while both patients followed here did display a similar overall pattern of disrupted functional pathways associated with their relapse events, there were notable differences. These differences would likely have been obscured if they had been part of a larger scale patient vs. control analysis. As personalised medicine is becoming more accessible, ME/CFS patients remain a patient group that will greatly benefit further from this style of investigation. Affected ME/CFS patients would be able to contribute to the overall understanding of the activity of their disease, and with individual molecular assessments be able to adopt therapeutic and behavioural management strategies that might better manage their illness and decrease the frequency of relapses during the long course of their disease as illustrated in Figure 6.

## 4. Materials and Methods

### 4.1. Cohort Recruitment

ME/CFS patients were recruited from Dunedin, New Zealand. Diagnosis was initially made by expert clinician, Dr Rosamund Vallings, of the Howick Health and Medical Centre, Auckland, NZ using the International Consensus Criteria [36]. The two patients and the healthy control. The two patients (aged 22 and 26 and the healthy control aged 24 were NZ European females of similar weight. Each was asked to self-report on their health status at each blood sampling indicating whether they were in a stable health period or in a more fragile or relapsed health state. Details of these assessments from each patient and control can be found in Figure 1A. The study conforms to the ethics approval 17/STH/188 for ME/CFS patient studies from the Southern Health and Disability Ethics Committee of New Zealand. General consultation with Ngai Tahu Research Committee of the University of Otago was carried out before the beginning of this research. 

### 4.2. PBMC Isolation

The study involved sampling of blood on 5 occasions from two patients and a healthy control over an 11-month period with the aim of catching a ‘relapse/relative recovery’ cycle of their illness. The patients filled out a brief survey detailing their current condition at the time of each blood collection. These brief health indicators are seen in Figure 1A. Blood was collected early to midmorning and the fractions were then processed within the same day. Peripheral Blood Mononuclear Cells (PBMCs) were isolated from the whole blood by layering on Ficoll-Paque before separating plasma from PBMCs and other cells by centrifuging at 400× *g*. The PBMC layer was pelleted (100× *g*) through PBS and the resulting pellet resuspended in PBS and RNA later and stored at −80 °C).

### 4.3. DNA Extraction

DNA was extracted from 200 μL of the PBMC fraction using the Illustra blood Genomic Prep Mini Spin Kit (GE Healthcare UK Ltd., UK) according to the manufacturer’s instructions. DNA was eluted into the provided EB buffer. Concentration was determined utilising the Qubit 2.0 fluorometer, following the Qubit dsDNA HS Assay Kit protocol (ThermoFisher Scientific, USA).

### 4.4. Generating Methylation Map Using RRBS

RRBS libraries were prepared as previously described [16,37,38]. Briefly, genomic DNA (500 ng) was digested with 160 U of MSP1 restriction enzyme (NEB, USA). Following end repair and adenylation of 3′ ends, adaptors were ligated to the DNA fragments. Bisulfite conversion was performed using the specifications of the EZ DNA methylation kit (Zymo research, USA). Semi-Quantitative PCR was performed on the bisulfite converted DNA in order to determine the optimal amplification cycle needed for the final large-scale PCR of the final library. Following PCR amplification of the DNA it was size selected using a 6% (*w*/*v*) NuSieve Gel (Lonza bioscience, USA) in order to extract the 40–220 bp desired fragments for RRBS libraries and to minimize adaptor contamination. Following purification and analysis of quality using a BioAnalyzer (Agilent, USA) and Qubit (Thermofisher Scientific USA) measures, samples were further purified using AMPure XP Bead (Beckman Coulter, USA) purification. 

### 4.5. High-Throughput Sequencing

The samples were sequenced through the Otago Genomics and Bioinformatics Facility (Dunedin, New Zealand). Following sequencing the raw fastq files were checked for adaptor presence and trimmed. The data were aligned to the human genome version GRCh37/hg19 using Bismark bowtie alignment generating BAM files utilised in the differential methylation analysis.

### 4.6. DNA Methylation and Statistical Analysis

Analysis was performed with the DMAP analysis program [39,40] run on a MAC OS X computer in order to investigate regions of methylation variability within each individual across fragments 40–220 bp in length. DMAP applied a Chi-squared test comparison for each individual. The fragment-based analysis approach has been well described previously [41,42,43]. All samples collected from each individual were included in these analyses. A minimum of two CpGs in each fragment had a minimum of at least 10 sequencing hits in order for the fragment to qualify. A Chi-square distribution test was performed on the five samples taken from each individual in this longitudinal study. False discovery rate corrected *p* values were calculated for each fragment and only fragments that met the significance threshold of FDR <0.05 were used in the remaining analysis. The genomic features overlapping with the fragments were identified using the DMAP Geneloc function.

Differential methylation was performed on each patient compared with the control producing gene lists, i.e., differentially methylated fragments directly overlapping with exon/intron regions. These gene lists were then analysed with pathway enrichment analyses using String.org [44]. A FDR *p* value cut-off of 0.05 was applied to select the enriched pathways. 

Fragments associated with patient relapse events were identified using 577 common ME-iVMFs detected across the three individuals. A fragment was associated with the relapse condition if it was found to have at least 15% average methylation difference between the relapse and recovery states, and if the methylation scores had a Pearson’s correlation coefficient of at least 0.9. In order to identify the functional associations of each variably methylated fragment associated with patient relapse events the regions covered by the ME-iVMFs of interest were investigated using the UCSC genome browser to compile a list of archived overlapping enhancers, promoters and regions of regulatory interactions. The associated genes were determined using the Genehancer database [45]. The functional roles of these genes were determined using Genecards [46] which was then used for determining appropriate functional categories for downstream analyses. 

## 5. Conclusions

This study shows the benefits of precision medicine for individual patients with a disease as physiologically complex as ME/CFS. Currently, ME/CFS patients can respond quite differently to specific medications, for example supplements like vitamin B12, and to anti-inflammatory drugs like naltrexone, and to physiological states like pregnancy, with some showing marked improvement, some marked deterioration, and some seemingly no change in their condition. By considering individual patients over the course of their ME/CFS disease we can better understand not only the similarities within the overall patient group, but also develop an in depth understanding of the fluctuations for each patient that relates to their specific pathophysiology. Variable methylation of regulatory regions associated with the relapse condition has in this study identified a number of genes with key functional roles in immune, inflammatory, metabolic and mitochondrial pathways. For a disease that has proven challenging to diagnose and characterise, with the delay in diagnosis detrimental for the affected person, this kind of analysis provides not only further evidence of serious biological dysfunction, but importantly also ongoing systematic molecular changes that inform future targets for individual treatment or symptom management as we continue to unravel and understand the complex nature of ME/CFS.

## Figures and Tables

**Figure 1 ijms-23-11852-f001:**
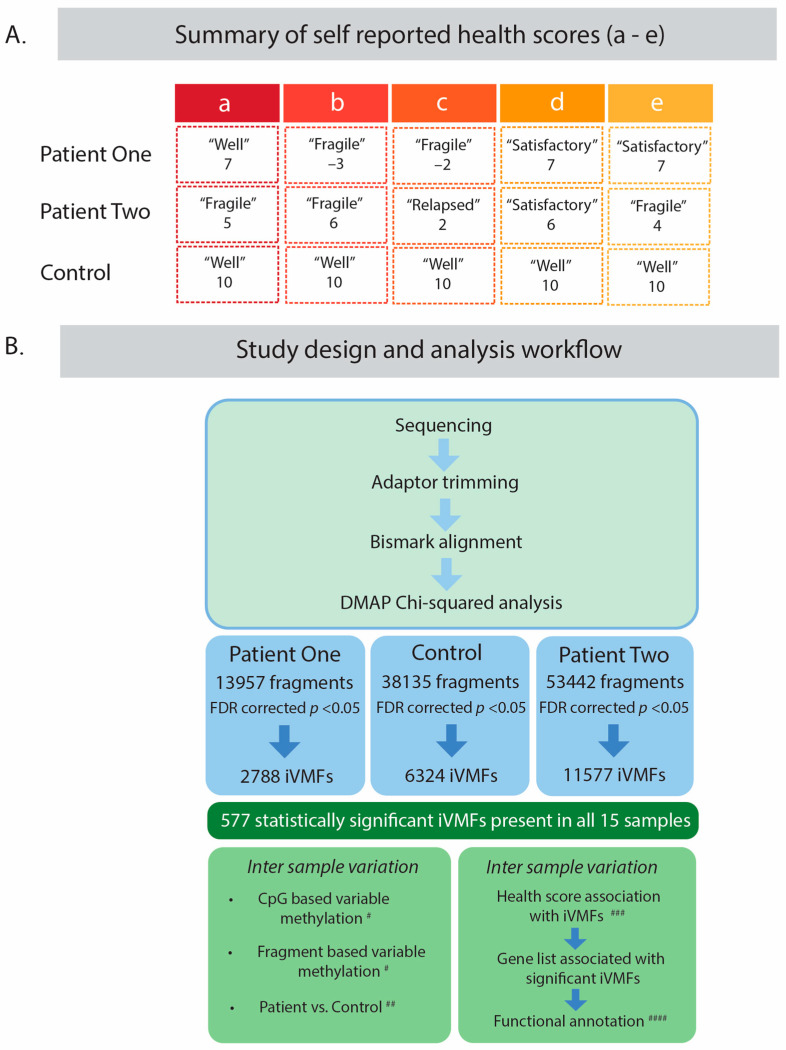
Study design. (**A**) Summary of self-reported health status of study subjects through a relapse and relative recovery cycle. The timeline of the health status of two patients over an 11 month period that spanned a relapse in each patient is shown along with the matched control. Self-reported information from the patients that they supplied on the day of blood donation indicated their health status (between –3 and 10). (**B**) The study design for longitudinal analysis. Following RRBS, and adaptor trimming and alignment of the data to human reference genome hg19 using Bismark, each sample was analysed. Initial estimates of variation utilised genome wide CpG methylation information, before the samples were analysed utilising the DMAP platform where a Chi squared analysis was used to identify methylation variation. The fragment methylation was also used to estimate variation but at key genomic locations. Comparisons were made between the patients and control. Continued analysis utilised the 577 statistically significant variable fragments identified across all 15 samples (FDR corrected *p* < 0.05) where correlation was calculated with health scores and ‘relapse’ and ‘recovery’ events to identify intra-individual variably methylated fragments (iVMFs). Genes associated with the individuals iVMFs were determined and functional categories investigated. # see Figure 2, ## see Figure 3, ### see Tables 1 and 2, #### see Figure 5.

**Figure 2 ijms-23-11852-f002:**
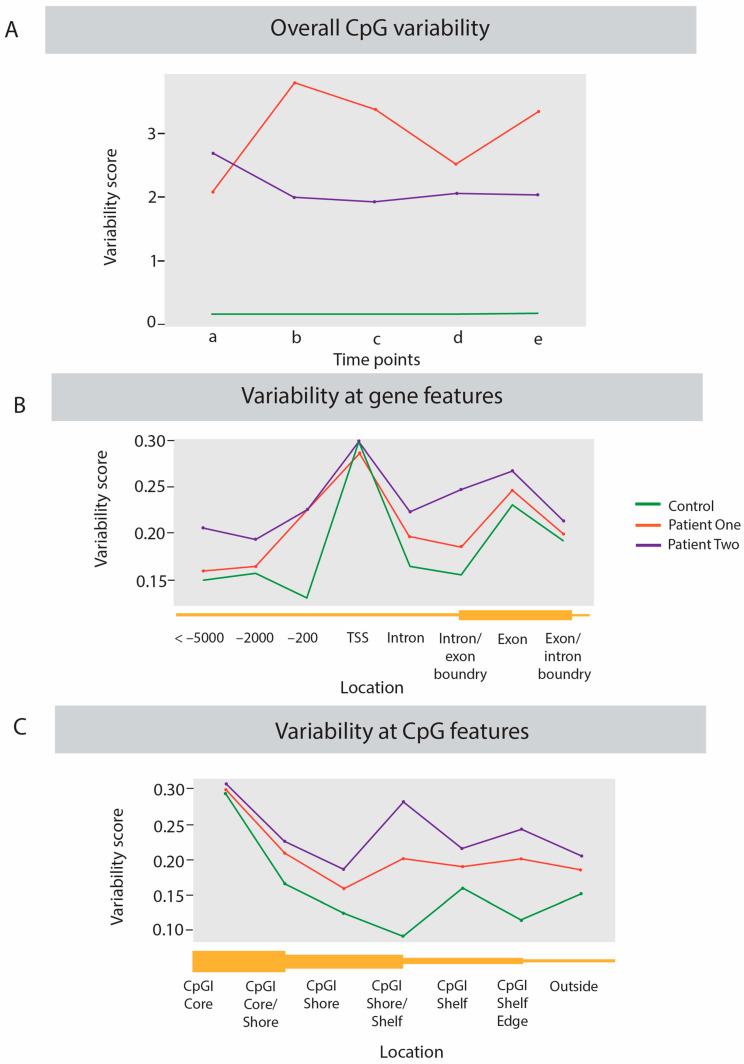
Line plots showing the variability calculated for each individual. (**A**) summary of the statistically significant unique differentially methylated CpG sites at each time point **a** to **e** for the two patients and the healthy control. The level of methylation variation was calculated utilising individual CpG methylation scores, with the significant differentially methylated CpGs found at that time point compared with the other time points, divided by the overall number of CpGs analysed for that individual, to give the percentage that were uniquely differentially methylated in each sample from each time points **a** to **e**. (**B**). Line plots showing the variability scores calculated for each individual across the gene features indicated on the x axis. Variability score was calculated by dividing the number of statistically significant (q < 0.05) variable fragments by the total number of fragments analysed at that feature. (**C**). Line plots showing the variability scores calculated for each individual across the features indicated on the x axis related to CpG Islands. CpG islands were defined as regions less than 500 bp with more than 55% GC content, CpG shores are defined as regions 2 Kb from the island with shelves 4 Kb away, the boundaries between these features are included (CpGI core/shore, shore/shelf and shelf edge). Variability score was calculated by dividing the number of statistically significant (q < 0.05) variable fragments by the total number of fragments analysed at that feature.

**Figure 3 ijms-23-11852-f003:**
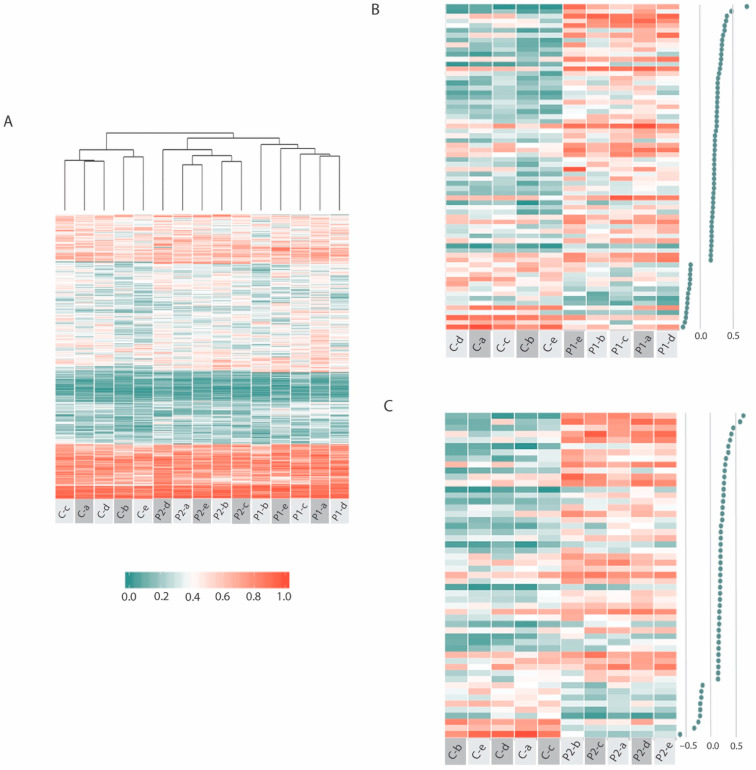
Heatmap showing the individual methylation variation at ME-iVMFs of interest. (**A**) Shows the methylation percent variation across all 577 statistically significant ME-iVMFs detected across all 15 samples. (**B**) shows the methylation variation in 68 ME-iVMFs where the mean methylation difference between the P1 and C groups is greater than 15% and (**C**) shows the methylation variation in 53 ME-iVMFs where the mean methylation difference between the P2 and C groups is greater than 15%. The dot plots associated with B and C on the right of the figure show the degree of differential methylation. The scale below Figure 3A shows the corresponding colour associated with the methylation scores.

**Figure 4 ijms-23-11852-f004:**
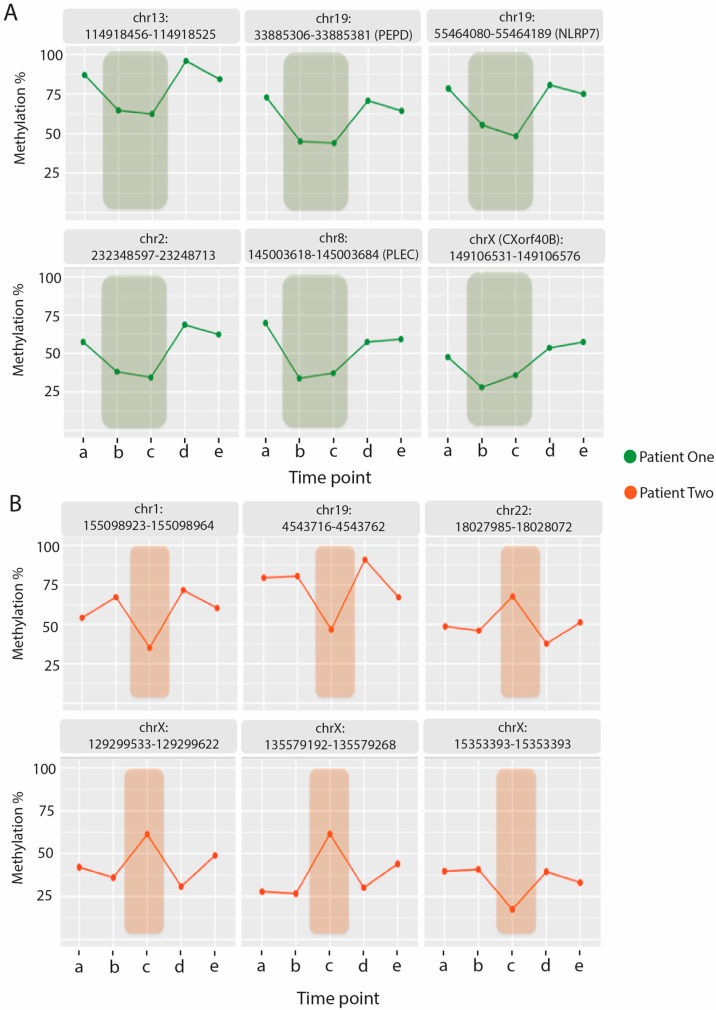
Dynamic DNA methylation variation correlated to self-reported patient health status. Methylation percentages are shown across all five time points **a** to **e** for each fragment of interest -fragment co-ordinates are shown above each block and a gene id is shown in brackets if the fragment directly overlaps a gene feature (intron/exon). (**A**) P1-green highlight indicates a period of relapse, (**B**) P2-orange highlight indicates period of relapse captured in one blood sampling.

**Figure 5 ijms-23-11852-f005:**
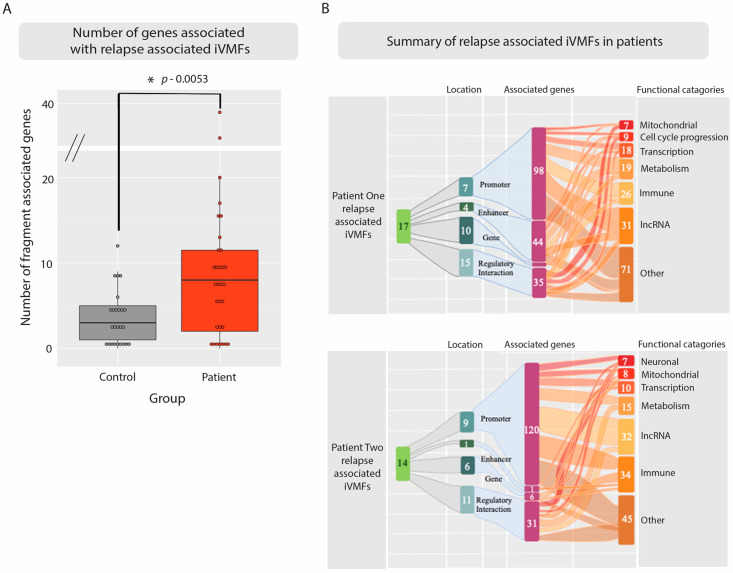
Investigation of the genes associated with relapse associated ME-iVMFs. (**A**) Box plots showing the number of genes linked with each statistically significant variably methylated fragment associated with a ‘simulated’ relapse event for the control in grey (in two analyses using each of the patient health relapse time points) and the ‘real relapse events’ of the two patients in red. Each point represents a fragment with the number of associated genes shown on the *y*-axis. The mean number of genes associated with the identified fragments for the patient and control groups is shown with the horizontal line. An unpaired *t*-test resulted in a significance value of *p* = 0.0053. (**B**) Sankey plot showing relationship between the variably methylated fragments identified in each patient associated with a relapse event and the biological functions they associate with through various regulatory genomic elements of relevant genes. From the statistically significant variably methylated fragments identified for each individual the location was determined and relevant regulatory interactions were recorded from UCSC genome browser. A gene list was compiled of genes associated with these regulatory interactions and the functional annotations were utilised to place them into categories. Some genes fell into multiple categories with others having no known function. Tables showing the full gene list, function and functional category is included in Appendix A ‘Genes_associated with P1’ and ‘Genes associated with P2’.

**Figure 6 ijms-23-11852-f006:**
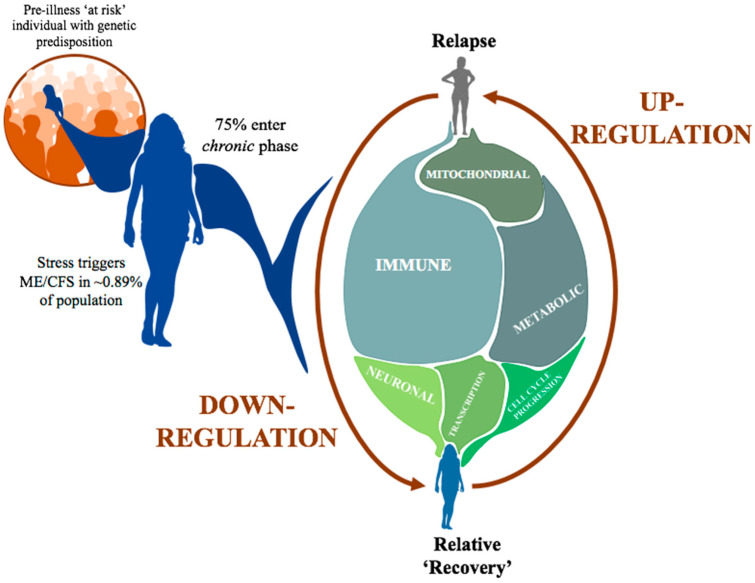
Summary of the longitudinal timeline of a ME/CFS patient. The initial external trigger for ME/CFS is a ‘stress event’ (for example, a viral infection- upper left in Figure 6) in a genetically susceptible person. Following progression into the chronic state patients experience frequent relapse events, which as this investigation suggests are primarily associated with the up regulation of a number of key biological systems.

**Table 1 ijms-23-11852-t001:** Fragments associated with the relapse condition for P1. Fragments with a Pearson’s correlation coefficient of at least 0.9 and a mean methylation difference between the “relapse” and “recovery” time points of at least 15% are shown. The table describes the location of each fragment, and the gene id if appropriate. It lists any overlapping regulatory elements (promoters/enhancers) recorded in Genehancer and the gene id associated with clusters of regulatory interaction determined using UCSC genome browser. Additionally, the table shows the methylation percent recorded for each time point (P1-**a** to P1-**e**) at each fragment with the relapse time points italicised.

Frag	Chr	Start	End	Position	GeneID	Genehancer	Regulatory Interactions	P1-a	*P1-b*	*P1-c*	P1-d	P1-e
1	19	33,885,306	33,885,381	On Intron	PEPD	GH09J033388	CEBPG:PEPD	78	*45*	*44*	71	64
2	X	150,565,438	150,565,527	On Intron	VMA21	GH0XJ151395	VMA21	50	*33*	*31*	45	49
3	19	55,464,080	55,464,189	On Intron	NLRP7	GH19J054952	NLRP2	78	*55*	*48*	81	75
4	7	5,741,705	5,741,780	On Intron	NF216	GH07J005687	ACTB:CCZ1:RNF216:USP42	86	*60*	*74*	84	88
5	X	135,579,269	135,579,310	On Intron	HTATSF1	-	-	39	*28*	*22*	46	40
6	X	152,908,188	152,908,279	On Intron	DUSP9	-	DUSP9	46	*18*	*30*	37	45
7	17	45,925,149	45,925,204	On Exon	SP6	GH17J047846	SP2:CDK5RAP3: OSBPL7:SCRN2	46	*28*	*29*	46	44
8	X	23,761,294	23,761,378	On Exon	ACOT9	GH0XJ023741	ACOT9	37	*18*	*18*	34	37
9	8	145,003,618	145,003,684	On Exon	PLEC	GH08J143914	ZC3H3:EEF1D:PLEC	70	*34*	*37*	58	59
10	X	149,106,531	149,106,576	On Exon	CXorf40B	GH0XJ149937	INC00B94:CXorf40B	48	*28*	*36*	54	58
11	22	42,316,243	42,316,306	Intergenic	-	GH22J041918	WBP2NL:CYP2D8P:CENPM:CYPSD6:TNFRSF13C	44	*30*	*27*	44	44
12	1	17,199,256	17,199,369	Intergenic	-	-	NECAP2:CROCC	55	*35*	*44*	59	57
13	2	232,348,597	232,348,713	Intergenic	-	-	NMUR1:NCL	57	*38*	*34*	69	62
14	13	114,918,456	114,918,525	Intergenic	-	-	CDC16:UPF31: RASA3	87	*65*	*62*	96	84
15	2	26,521,360	26,521,433	Intergenic	-	GH02J026298	HADHB:HADHA: ADGRF3	53	*38*	*35*	60	53
16	3	10,334,731	10,334,778	Intergenic	-	GH03J010291	GHRLOS:GHRL	33	*13*	*22*	40	41
17	15	22,095,431	22,095,475	Intergenic	-	-	-	51	*40*	*34*	51	56

**Table 2 ijms-23-11852-t002:** Fragments associated with the relapse condition for P2. Fragments with a Pearson’s correlation coefficient of at least 0.9 and a mean methylation difference between the “relapse” and “recovery” time points of at least 15% are shown. The table describes the location of each fragment and the gene id if appropriate. It lists any overlapping regulatory elements (promoters/enhancers) recorded in Genehancer and the gene id associated with clusters of regulatory interaction determined using UCSC genome browser. Additionally, the table shows the methylation percent recorded for each time point (P2-a–P2-e) at each fragment with the relapse time point italicised. Fragments hypermethylated in the relapse condition are shown in bold.

Frag	Chr	Start	End	Location	GeneID	Genehancer	Regulatory Interactions	P2-a	P2-b	P2-c	P2-d	P2-e
1	19	4,543,716	4,543,762	On Intron	SEMA6B	GH19J004539	YJU2:PLIN5:SEMA6B:LRG1	80	81	*47*	91	67
2	X	15,353,393	15,353,501	On Intron	PIGA	GH0XJ015333	ZRSR2:PIGA	40	41	*18*	39	33
3	22	17,640,812	17,640,923	On Intron	CECR5	GH22J017157	HDHD5	69	82	*60*	80	71
4	14	105,936,238	105,936,292	On Exon	MTA1	GH14J105464	IGHGP:CDCA4:CRIP2:MTA1:ENSG00000257270	84	80	*63*	87	76
5	22	18,027,985	18,028,072	On Exon	CECR2	-	-	49	46	*68*	38	51
6	X	102,565,776	102,565,848	Intron–Exon Boundary	BEX2	GH0XJ103310	BEX2	42	45	*26*	43	41
7	1	155,098,923	155,098,964	Intergenic	-	GH01J155123	DAP3:CLK2:DPM3:GBAP1:THBS3:EFNA1	54	67	*35*	72	60
8	3	10,334,731	10,334,778	Intergenic	-	GH03J010291	GHRLOS:GHRL	49	56	*32*	61	41
9	9	38,687,682	38,687,760	Intergenic	-	-	-	61	62	*39*	65	42
10	7	100,882,140	100,882,220	Intergenic	-	GH07J101231	FIS1:CLDN15	80	80	*65*	90	75
11	2	219,233,608	219,233,704	Intergenic	-	GH02J218366	AAMP:SCL11A1:TMBIM1:CATIP	49	55	*38*	62	52
**12**	**6**	**170,403,979**	**170,404,085**	**Intergenic**	**-**	**-**	**WDR27**	**62**	**66**	** *81* **	**63**	**70**
**13**	**X**	**129,299,533**	**129,299,622**	**Intergenic**	**-**	**GH0XJ130164**	**ELF4:AIMF1:ZNF280C**	**42**	**36**	** *61* **	**31**	**49**
**14**	**X**	**135,579,192**	**135,579,268**	**Intergenic**	**-**	**-**	**-**	**28**	**27**	** *61* **	**30**	**44**

## Data Availability

All datasets generated and analysed during this current study are available in the GEO database NCBI (GSE166592).

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
