# Peer review of "Dynamic Epigenetic Changes during a Relapse and Recovery Cycle in Myalgic Encephalomyelitis/Chronic Fatigue Syndrome"

_ijms, 2022, doi:10.3390/ijms231911852_

Round 1
Reviewer 1 Report
The manuscript investigated dynamic epigenetic changes specific to each individual between ME/CFS patients and healthy controls mapping genomic changes in selected ME/CFS patients through a relapse recovery cycle. Analyses were performer on genomic DNA isolated from two patients and a healthy age/gender matched control to capture a patient relapse.
In my opinion, the conclusion that severe health relapses in ME / CFS patients result in functionally important changes in DNA methylomes which, while differing among patients, lead to similar compromised physiology would it require more individuals examined and I can imagine that a greater numbers of patients and controls were analyzed but it does not seem to me that this is mentioned in the submitted manuscript.
I don’t understand, in fact, at lines 590-591 "The two patients and the healthy control were NZ European females (23-28 years of age) of similar weight. It is not clear what is the age of the patients and what is the age of the only healthy control, I believe that the authors should specify this point better and possibly add or mention the results of analyses derived from at least a second healthy control.
I personally find this work interesting, well designed and fairly well documented. The manuscript is clear, and presented in a well-structured manner.
In my opinion the methodological approach (dynamic study of change of methylation pattern) could open the way to other studies in the field and provide an advancement of the current knowledge in the field but, also in those of nutriepigenomics, nutritional epidemiology, etc. In fact, it would be interesting to consider also the effect of the diet to interpret the results but the Authors do not mention this aspect in the presented manuscript (ijms-1909319).
In particular, my question is both the two ME/CFS patients and the healthy control have followed the same dietary regimen? It will be possible verify whether some foods improve/worse the severity of symptoms in correlation with the dynamic analysis of DNA methylation as reported in the paper submitted for publication?
I believe that the results provide an advancement of the current knowledge and the publication of this manuscript, if revised, could encourage more work in the field. For this reason, I would not be unfavorable to publication if the Authors integrated data obtained from at least two controls, also mentioning the diet. There are in fact several recent published works that demonstrate for example how Dietary Factors Affect DNA Methylation (e.g., Maugeri and Barchitta, 2020 etc.)
I also suggest to check the text format and font style (e.g. example at line 669-670-6).
Author Response
Response to reviewer 1
greater numbers of patients and controls were analyzed?
Our previous studies (transcriptome, proteome, methylome etc ) with our ME/CFS cohorts have been single time point studies of ME/CFS vs healthy controls and have included a larger cohort of patients, and age gender matched controls for each patient. They depended on data from the healthy controls to provide the control data for the ME/CFS patients so we could identify changes within the ME/CFS cohort. These studies therefore provided a molecular “snapshot” of a fixed time of sampling.
Here, we have taken a different approach to evaluate individual patients in longitudinal studies where they are their own control. Our rationale in this study is to discover dynamic changes over a time period that are associated with health status of ME/CFS patients. We wanted to provide a proof of principle of dynamic methylation changes in individuals. By following the patient over a year and capturing a relapse and relative recovery phase of their ME/CFS health, we are comparing the methylome data at the time of the relapse, with the data before and after relative recovery in each of the patients studies. That forms the control data. We were excited that in each patient we could see specific methylome changes occurring during the relapse compared with before and they were restored on relative recovery. This study provided the evidence (for the first time in ME/CFS as per our knowledge) that methylation changes could potentially be used in personalised epigenetic context to track relapse/recover cycle which was our main rationale as indicated above. This study demonstrated valuable information can be obtained at the single patient level. Both patients had the expected health changes characteristic of a classic ME/CFS relapse and that was reflected in very similar effects on specific physiological pathways during the relapse , although there were individual methylation differences at different loci between the two patients leading to the same pathways being affected. This observation clearly provides rationale and validity of our personalised epigenetics study, where different methylation sites are affected in the individuals, however, these changes led to effect the same biological pathway.
We added a healthy control so we could evaluate the changes throughout the sampling period to an individual who had no health relapse. As with the patients, the longitudinal data formed its own control data. We did simulate a ‘theoretical relapse’ in the same times of the year when the patients suffered relapses and determined there were not specific methylation changes leading to the same altered physiology as we found for the patients (variation was within the healthy control’s low background variation from month to month).
So for the purpose of this particular study we feel there is no added value in including another control. Also note this study is not a case-control study where higher number of cases or control will incrementally add power to the study. This is personalised epigenetics study attempting to reveal changes in patients over time (by multiple sampling at different times) that are linked with their health status in the context of ME/CFS. Obviously in follow up studies we would want to evaluate larger cohorts with more individual patients and controls and examine how their DNA methylomes vary during a relapse and see whether there are subtypes within the physiological dysfunction.
Lines 590-591 not expressed clearly
We have rephrased this sentence to make it clearer and given the ages when the blood was donated. The two patients at the time of the study were 21 and 26 respectively, and the healthy control 24, so the age range was actually 21-26 .
Were both the two ME/CFS patients and the healthy control following the same dietary regimen
These two patients have taken part in many of our studies. They had been diagnosed by the same expert ME clinician and we had detailed information on their general lifestyle, including their diets, and level of activity. They were carefully selected to be as closely matched as possible including age and gender and that their diets were similar and healthily balanced with the main nutritional groups. So in this case dietary differences should not have contributed significantly to the and differences in the data.
However, the healthy control was a very active young woman who had a similar dietary regime, but obviously her exercise and general activity was much higher than the two patients with ME/CFS who had limited activity to avoid post exertional malaise. Indeed, in a separate and ongoing study on post exertional malaise involving the three young women have documented their relative abilities to exercise and the effects on their cardiac physiology. In this case the different levels of physical activity may have contributed to the differences in stability of the methylome between the healthy control and two patients. This will be a subject of our future investigation.
I have added this information and 2 new references on diet and lifestyle and the epigenetic code to the manuscript and appreciate the point the reviewer raised on diet and that that it was important to communicate this information.
I have also added two further references to the manuscript on our groups use of the methodology to strengthen the referencing.
Text format and font style line 669-670-6 :
This seems to be correct
Reviewer 2 Report
Thank you for the opportunity to review the manuscript on dynamic epigenetic changes during a relapse and recovery cycle in ME/CFS. Overall the paper is well written and the figures are well made and informative. In this study, the authors evaluate genomic changes in two female patients with ME/CFS and one female healthy control over a period of 11 months (5 blood draws). The authors found that during relapses, the two patients with ME/CFS had DNA methylation changes that indicate disturbed metabolic, immune, and inflammatory functions. The tone of the paper is descriptive and concludes that patients with ME/CFS have complex physiology. The research design is appropriate for this exploratory study but future studies by the authors should focus on expanding the number of cohorts.
Major Comments:
1) The main limitation of this study is the small patient sample size (2 patients with ME/CFS and only one healthy control, all participants are female) and the limited information about the cohorts (amount of time with ME/CFS diagnosis, medications that patients are taking, comorbid medical conditions, illnesses around the time of blood collection, pregnancy or prior pregnancy status, dietary habits, etc). These limitations should be addressed in the discussion.
Author Response
Response: The reviewer notes the small size of the study.
Our previous studies ( transcriptome, proteome , methylome etc ) with our ME/CFS cohorts have been single time point studies of ME/CFS vs healthy controls and have included a larger cohort of patients , and age gender matched controls for each patient. They depended on data from the healthy controls to provide the control data for the ME/CFS patients so we could identify changes within the ME/CFS cohort. These studies therefore provided a molecular “snapshot” of a fixed time of sampling.
Here, we have taken a different approach to evaluate individual patients in longitudinal studies where they are their own control. Our rationale in this study is to discover dynamic changes over a time period that are associated with health status of ME/CFS patients. We wanted to provide a proof of principle of dynamic methylation changes in individuals. By following the patient over a year and capturing a relapse and relative recovery phase of their ME/CFS health, we are comparing the methylome data at the time of the relapse, with the data before and after relative recovery in each of the patients studies. That forms the control data. We were excited that in each patient we could see specific methylome changes occurring during the relapse compared with before and they were restored on relative recovery. This study provided the evidence (for the first time in ME/CFS as per our knowledge) that methylation changes could potentially be used in personalised epigenetic context to track relapse/recover cycle which was our main rationale as indicated above. This study demonstrated valuable information can be obtained at the single patient level. Both patients had the expected health changes characteristic of a classic ME/CFS relapse and that was reflected in very similar effects on specific physiological pathways during the relapse , although there were individual methylation differences at different loci between the two patients leading to the same pathways being affected. This observation clearly provides rationale and validity of our personalised epigenetics study, where different methylation sites are affected in the individuals, however, these changes led to effect the same biological pathway.
We added a healthy control so we could evaluate the changes throughout the sampling period when as expected there was no health relapse. As for the patients, the longitudinal data formed the control data. We did simulate in the same times of the year when the patients suffered relapses and determined there were not specific methylation changes leading to the same altered physiology as we found for the patients (within the healthy control’s low background variation from month to month.
Limited information of the subjects
We acknowledge we did not discuss in detail information on the patients and have added more information to address the concerns of the reviewer.
These patients had taken part in many of our studies, had been diagnosed by the same expert ME clinician and we had detailed information on co morbidities, medications, and general lifestyle, family status , including their diets, and level of activity. They were carefully selected to be as closely matched as possible including age and gender and that their diets were similar and healthily balanced with the main nutritional groups. Indeed they had no comorbidities, had not experienced a pregnancy, both had taken melatonin to help with sleep . They both were paediatric cases have contracted the illness in their teens and had had their illness formally diagnosed for 6 and 10 years respectively but it is likely to have started earlier, .
I have added this information in the manuscript and thank the reviewer for reminding us that it was important to communicate this information.
Round 2
Reviewer 1 Report
I express a favorable opinion for the publication of your study because In my opinion the methodological approach (dynamic study of change of methylation pattern) could open the way to other studies in the field and provide an advancement of the current knowledge in this field.
I personally find new version of your work interesting, well designed and fairly well documented.
I believe that some aspects are now clearer than in the previous version of your work and It will be interesting to read about the results of your future investigation about role of physical activity on stability of human methylome.